# How Artificial Intelligence in Imaging Can Better Serve Patients with Bronchial and Parenchymal Lung Diseases?

**DOI:** 10.3390/jpm12091429

**Published:** 2022-08-31

**Authors:** Trieu-Nghi Hoang-Thi, Guillaume Chassagnon, Hai-Dang Tran, Nhat-Nam Le-Dong, Anh Tuan Dinh-Xuan, Marie-Pierre Revel

**Affiliations:** 1Department of Diagnostic Imaging, Vinmec Healthcare System, Ho Chi Minh City 70000, Vietnam; 2AP-HP. Centre, Cochin Hospital, Department of Radiology, Université de Paris, 75005 Paris, France; 3AP-HP. Centre, Cochin Hospital, Department of Respiratory Physiology, Université de Paris, 75005 Paris, France

**Keywords:** artificial intelligence, bronchial disease, parenchymal lung disease, machine learning

## Abstract

With the rapid development of computing today, artificial intelligence has become an essential part of everyday life, with medicine and lung health being no exception. Big data-based scientific research does not mean simply gathering a large amount of data and letting the machines do the work by themselves. Instead, scientists need to identify problems whose solution will have a positive impact on patients’ care. In this review, we will discuss the role of artificial intelligence from both physiological and anatomical standpoints, starting with automatic quantitative assessment of anatomical structures using lung imaging and considering disease detection and prognosis estimation based on machine learning. The evaluation of current strengths and limitations will allow us to have a broader view for future developments.

## 1. Introduction

Artificial intelligence (AI) is defined by the ability of a computer system to complete a task without direct intervention or supervision by human intelligence [1]. This field of science was founded on the assumption that “that every aspect of learning or any other feature of intelligence can in principle be so precisely described that a machine can be made to simulate it” [2].

Machine learning is one of the main AI techniques for improving the intelligence of machines (computers) by providing them with experience-based learning. An artificial neural network (ANN) is a learning technology that models information in a similar manner to neuronal cells network of living organisms. Deep learning is behind the current rise of AI, thanks to the juxtaposition of many neural networks and the rise in computing power. “Artificial perception” is defined as when AI mimics the way humans use their sensory organs to interact with their environment. If those senses are visual, it is called “computer vision.” In the field of radiology, there are many applications, such as detection tasks also called computer-aided diagnosis (CAD) or classification tasks. Classification tasks can be based on a process called radiomics, which consists of selecting imaging parameters of varying complexity, that cannot be caught by the human eye, and can enable establishment of correlations with clinical findings (Figure 1).

When looking back on the history of AI, one realizes the importance of the Dartmouth conference in the summer of 1956 as this was undoubtedly the founding event of AI as a scientific field on its own [3], one year after John McCarthy had chosen the term “artificial intelligence” in part for its neutrality [2]. Thus, AI was born, but it was not the result of a single event; as is the case for most historical events, it resulted from the assimilation of many previous initiatives.

In the following years, AI has generated several waves of optimism [3,4], mixed with disappointments and loss of funding [5,6], followed by new approaches, successes, and renewed funding (Figure 2) [7]. Medicine was not excluded from this development, particularly in the field of disease detection and the creation of CAD tools. Advances in research on CAD systems are well aligned with advances in AI. Figure 2 illustrates this relationship in which the x- and y-axes represent the timeline by year and the level of expectation for AI, respectively.

CAD tools in medicine have evolved over time since 1968, when research officially began The rapid growth of CAD research began with automated CAD in the 1980s and culminated in the first commercial CAD system for mammography, the ‘Image Checker’ system, approved in 1998 by the United States Food and Drug Administration [8]. With the third AI boom, it is expected that CADs [8] will adapt and then improve their performance in response to clinical problems and limitations of contemporary technology. Thus, in the 21st century, AI techniques staged a remarkable comeback with the simultaneous developments and advances in computational power, big data, and theoretical approaches. AI techniques have since become an essential part of the industry, that has integrated this tool to solve many challenging problems in various fields including computer science, software engineering, and operations research.

## 2. Artificial Intelligence Applied in Computer Tomography Thoracic Imaging in the Scope of Bronchial and Parenchymal Lung Diseases

In this part of the review, we try to synthesize the technical principles that have been used in the study of bronchial and lung parenchyma pathologies along with certain achievements (Figure 3). We did not mention in this part the concerns of oncology which is beyond the scope of this review.

### 2.1. Density Measurements

Low density area. Zones of low density (LDZ) are an index of lung density, defined as the percentage of parenchymal voxels below a given X-ray attenuation threshold. The −950 Hounsfield unit (HU) is the most widely used [9,10] to quantify emphysema and has been validated by histology. Quantitative assessment of emphysema using this threshold has better reproducibility than visual assessment

High density area. Zones of high density (HDZ) are an index of lung density, defined as the percentage of attenuation parenchyma voxels between −600 and −250 HU. ZHD are associated with smoking [11] and a higher risk of interstitial lung disease [12]. HDZ is associated with inflammatory biomarkers, extracellular matrix remodeling, impaired lung function and a higher risk of death [13].

Recently, automatic parenchyma quantification by density thresholding has been applied in the evaluation of pneumonia due to COVID-19: emphysema for thresholds from −1024 HU to −977 HU [14], residual healthy parenchyma from −977 HU to −703 HU [15,16], ground glass from −703 HU–−368 HU and consolidation from −100 HU to 5 HU [15,17]. The group of Roberto et al. showed that COVID-19 pneumonia can be evidenced by a median ground glass percentage value of 19.50% and that ground glass volume alone showed significant difference between patients with suspected COVID-19 pulmonary infection as compared to controls.

Air trapping, defined by excessive respiratory gas retention in abnormal parts of the lung during expiratory maneuvers, can be represented by the percentage of LDZ on expiratory CT images. It can be calculated for the whole lung, as well as for a lung segment or sub-segment. Different thresholds have been proposed to quantify air trapping, among which are −856HU [18], −850 HU [19] or −900 HU [20]. The −856 HU threshold is the most widely used. Air trapping measurement on CT is significantly correlated with lung volumes and small airways expiratory flow measurements in COPD [21,22] and asthma [18,23], and can be used to assess treatment response [24].

### 2.2. Histogram Analysis

Global density histogram parameters of CT images—e.g., asymmetry, kurtosis, and mean lung density—are useful in estimating the extent of interstitial lung disease—ILD [25,26]. For example, collagen deposition increases lung density in pulmonary fibrosis, which in turn causes a rightward shift of the histogram frequency and reduces its peak, thereby increasing asymmetry and kurtosis [27]. These parameters are reproductible [28] and they are not affected by radiation dose reduction in patients with idiopathic interstitial pneumonias [29].

Kurtosis and the visual extent of the fibrotic profile score are the only indices predicting mortality in a retrospective study of 167 subjects with idiopathic pulmonary fibrosis—IPF [30]. It has also been recently shown that histogram parameters correlated with pulmonary function and they were associated with lung transplant-free survival in IPF patients in a comparable manner to visual assessment by two experts [31]. Histogram parameters can also distinguish between different well-defined mortality risk categories in subjects with ILD-associated systemic sclerosis (SSc) [32].

### 2.3. Texture Analysis

Currently, there are several software packages available for lung parenchyma texture analysis, each of which is capable of extrapolating and evaluating different groups of radiomics parameters (for examples CALIPER [33]). Textural analysis is based on regions of interest—ROI—in the lung selected by trained experts, according to a set of specific patterns (normal, reticulation, honeycombing, etc…). The histogram and texture parameters of each volumetric ROI are extracted to allow development of predictive models for specific “textures” using machine learning [34,35]. Most of the work on the detection of pulmonary fibrosis patterns has been based on the classification of 2D images using a textural patch-based approach involving division of the lung into small patches with similar size (e.g., 32 × 32 pixels) and inclusion of these patches into one of the classes of fibrotic patterns. These classifiers are then trained on databases comprising thousands of annotated patches, with representatives of each class able to differentiate ground glass from honeycombing and/or emphysema. However, this type of analysis approach is limited by an intrinsic subjectivity (prior training of the experts), and by the risk of misclassifying central or peripheral parts of the peripheral lung next to the chest wall. Interfaces between two classes of abnormalities can also be overlooked when the information related to the surrounding lung is not included in the patches [36,37]. There are other approaches such as segmentation of the fibrotic lung without separate quantification of each component (i.e., ground glass, reticulations, honeycombing) [38]. This requires contouring the abnormal fibrous areas on each CT scan image, which is time consuming but then allows the model to be applied to the entire lung.

### 2.4. Lung Shrinkage Detection

Elastic registration is used to quantify lung shrinkage in patients with lung fibrosis follow-up. CT images were elastically registered to match baseline images from which deformation maps can be generated. Elastic registration was performed using multimeric, multimodal graph-based registration algorithms [39]. Jacobian maps were derived from the logarithm of the Jacobian (log_jac) determinant for each voxel of the deformation matrix. The Jacobian determinant is quantitatively defined as the deformation (stretching or shrinkage) of each voxel, to generate baseline lung scan matching for the follow-up examination. Log_jac equates 0 when the voxel size remains similar after deformation. Log_jac is negative when the deformed voxel is smaller than the original; Log_jac is positive when the deformed voxel is larger [40]. Jacobian maps demonstrated lung parenchyma shrinkage of the posterior lung bases in patients with worsened ILD at visual assessment. Morphologic and functional worsening can be detected with an accuracy of 80% (32 of 40 patients; 95% confidence interval [CI]: 64%, 91%) and 83% (33 of 40 patients; 95% CI: 67%, 93%), respectively. Jacobian values significantly correlated with changes in lung function including forced vital capacity (R = 20.38; 95% CI: 20.25, 20.49; P, 0.001) and lung diffusing capacity for carbon monoxide (R = 20.42; 95% CI: 20.27, 20.54; P, 0.001).

### 2.5. Disease Extension Contouring

Automatic assessment of the extent of ILD-related SSc on chest CT images from patients with no, mild, or severe lung disease was developed using a multicomponent deep neural network (AtlasNet) and trained on fully annotated CT images. ILD contours as assessed by three readers and the deep learning neural network were compared using the Dice similarity coefficient (DSC) [41]. The median DSCs between the readers and the deep learning ILD contours ranged from 0.74 to 0.75, whereas the median DSCs between contours from the three radiologists ranged from 0.68 to 0.71. Less difference was therefore obtained between the algorithm and each of the readers than between the readers themselves. The disease extent correlations with lung diffusing capacity for carbon monoxide, total lung capacity, and forced vital capacity were r = −0.65, −0.70, and −0.57, respectively, in the external validation dataset (P, 0.001 for all) [42].

## 3. Strengths and Limitation of Artificial Intelligence in Thoracic Imaging

The field of medical imaging is currently making great advances in machine learning for public health care. In September 2019, GE Healthcare had FDA approval for marketing a set of algorithms aimed at detecting pneumothorax on chest X-rays [43]. An abstract presented almost concomitantly at the International Association for the Study of Lung Cancer (IASLC) provided evidence suggesting that CAD using a low-dose CT scanner is effective in screening lung cancer [44]. More surprisingly, a neural network generated by Google scientists had the same accuracy in detecting malignant lung nodules as the one provided by radiologists [45]. Another study published in 2018 reached the same conclusion, stating that deep-learning algorithms yielded the same efficiency as chest radiologists in classifying ILD [46]. Similar results were obtained when AI was applied to diagnose COPD in smokers and predict episodes of exacerbations and mortality [47].

Although many studies have focused on deep-learning methods applied to CT scans, a few have also investigated the added values provided by AI to the ‘classical’ chest X-ray platform [48]. The high accuracy in pneumonia detection and classification using deep-learning and texture analysis [49,50] have been recently applied to detect COVID-19 [51,52]. In May 2022, a quality review synthesized 40 articles on advances in applying machine learning in the diagnosis of pneumoconiosis in the coal industry [53]. Making the best use of AI will optimize physicians’ time in favor of patients’ care, but will also improve both diagnostic accuracy and therapeutic efficiency, thereby optimizing workflow. For example, using parameters based on density and texture, CAD can automatically identify abnormal chest radiographs in the radiologist’s worklist, thus reducing the turnaround time for reporting an abnormal X-ray by 44% [54]. CAD tools can be used for specific detection tasks on chest radiography, such as detection of tuberculosis, pneumonia, or pulmonary nodules [55,56]. Other even more advanced tasks such as the detection of multiple diseases are also under development. Huge insights into the impact of AI applied to chest imaging can benefit locations with a scarcity of experts able to interpret chest X-rays, even when imaging equipment and facilities are readily accessible [55,57].

There are, however, many challenges when applying deep-learning algorithms to medical image analysis. First, lack of large training databases is an unavoidable barrier in the very early period. In deep learning, having a large training database is essential according to the principle requiring larger number of images to improve training accuracy. Consequently, weak algorithms based on large quantity of data can be more accurate than strong algorithms derived from a small dataset [58]. Class imbalance, i.e., number of samples from one class being much higher than from the other, represents another serious bias [59]. Deep-learning algorithms accuracy therefore critically depends on the fact that the number of samples in each class must be kept equal or balanced, a task that cannot always be achieved. To circumvent this difficulty, one can enlarge the training database without obtaining new images by using image augmentation techniques and generating variations of the original images. This can be achieved by implementing different processing methods using as rotations, flips, translations, zooms and adding noise to the original image [60].

Transfer learning, a popular method in computer vision, provides another strategy to address the data restriction problem [61]. Transfer learning allows reuse of a model learned from a given domain to a different one. Transfer learning can be performed with or without a pre-trained model. The pre-trained model is a model developed to solve new tasks which are very similar to the previous ones. This approach can save time as it can re-use existing models as a starting point. For example, many ConvNet architectures are pre-trained on ImageNet [62]. Bush et al. pretrained a ConvNet model on a subset of ImageNet data and it was able to label chest radiographs as positive or negative for the presence of lung nodules with a sensitivity of 92% and a specificity of 86% [63].

Currently, PACS systems, which are fed by millions of images, have been commonly used in radiology in most Western hospitals for at least a decade. More and more public databases are made available to researchers, e.g., those evaluating image analysis algorithms to detect lung nodules in lung CT from LIDC-IDRI [64], or data from the cohort of patients tested for COVID-19 at SUNY [65]. One of the largest databases is ChestX-ray8, built from the clinical PACS of hospitals affiliated with the National Institute of Health. This database includes 111,220 frontal view radiographs of 30,805 patients and labelled images of 8 diseases, then expanded to 14 diseases (chestX-ray14) including atelectasis, consolidation, infiltration, pneumothorax, edema, emphysema, fibrosis, effusion, pneumonia, pleural thickening, cardiomegaly, nodule, mass and hernia [66].

Another challenge concerns the acquisition of relevant annotations or labeling for medical images. Given the complexity of training algorithms in healthcare, researchers typically need the advice of domain experts (e.g., radiologists, pathologists) to make task-specific annotations for image data. This type of supervised analysis is limited by built-in subjectivity, and without proper quality control, machine learning might erroneously replicate human misjudgments. These reasons explain the current renewed interest in unsupervised learning by the machine learning community. Unsupervised methods are attractive because they allow the initial training of the network using the abundance of unlabeled data available worldwide [67].

As CT images are by definition 3D, using ConvNet is more complex than for 2D chest X-ray images. Labeling a large database is time consuming, and thus problematic. For example, slice-by-slice annotations are required to train deep-learning systems for 3D CT image segmentation, but this is a challenging process due to the high number of slices. To overcome this problem, one can use ConvNets 2D applied to each slice, or follow a patch-based approach and reduce image size, with the risk of losing information in both cases. The results are nevertheless promising and ConvNets neural networks generally achieve better analytical outputs than traditional machine learning methods. For example, the Caliper software uses the patch-based approach; Caliper software designed to quantify disease extent and progression in IPF patients is therefore currently gaining momentum in AI clinical applications [68,69].

Images from Chest CT are usually obtained from two different reconstructions, using high frequency and pulmonary kernels for the evaluation of the lung parenchyma whilst standard reconstruction and mediastinal kernels are used to evaluate the mediastinum. Recent studies suggest that the performance of deep-learning based tools for the detection and characterization of lung nodules is influenced by acquisition and reconstruction parameters, such as the choice of reconstruction kernel [70,71]. Current data evaluating the influence of reconstruction filters on lung disease segmentation using data-driven research methods demonstrated that training on both mediastinal and lung kernel images can improve the performance of the model [72].

In medical image analysis, useful information does not only stem from the images themselves. Additional data on patient histories, age, demographics, and other paraclinical exams are also useful to improve diagnostic precision. One can assess the combination of this information in deep-learning networks in a simple way [73]. The improvements achieved were, however, not as great as expected, and the current challenges include achieving proper balance between the large number (usually thousands) of imaging features used in deep-learning networks with the very small number of clinical features to prevent neglect of the latter.

Just as human decision-making processes may rely on “instinct”, that cannot be clearly explained or deciphered, neural networks can sometimes also be considered as “black boxes”. One must take this into consideration, especially in situations where liability is important and can have serious legal consequences. In other words, having a good prediction system might not be sufficient for dealings with the public. Consequently, several strategies have been developed to better understand the impressive and complicated maze represented by the middle layers of convolutional networks, including deconvolution networks [74], guided backpropagation [75], or deep Taylor decomposition [76] with the hope to attenuate the public perception of AI as “black box” mechanisms.

Finally, as for all clinical applications, artificial neural networks must be extensively trained and validated. As described above, there is, however, an imbalance between the amount of published technical innovations, and the actual number of clinical applications. One of the reasons is publication bias, related to the lack of external validation, or very specific databases stemming from one center reported in many publications. Although carrying out validation studies is, by itself, a limiting factor, and there are numerous administrative hurdles to overcome to obtain clearance from regulatory agencies, clinical investigators are encouraged to concentrate their efforts to embed AI into daily clinical practice. In the meantime, software validation is now mainly carried out by industrial companies, prioritizing profitable financial rather than scientific targets. For this reason, closer collaboration between academic, including technical, biological, and clinical researchers, and industrial partners are needed. For these types of collaborations and to fill this gap, interdisciplinary research at universities is essential [77]. Initiatives from different professional organizations launching new training programs for the use of AI in medical imaging are therefore to be welcomed.

## 4. Conclusions

The above are our conclusions about artificial intelligence applied to lung imaging in patients with bronchial and parenchymal lung disorders. This new approach will improve personalized approaches to both diagnostic performance and therapeutic efficiency and lead to better medical outcomes. At a time when human health is threatened by emergent and deadly pandemics affecting millions of individuals, e.g., COVID-19, accurate means to rapidly identify at risk patients with preexisting lung comorbidities will reduce global mortality rates.

## Figures and Tables

**Figure 1 jpm-12-01429-f001:**
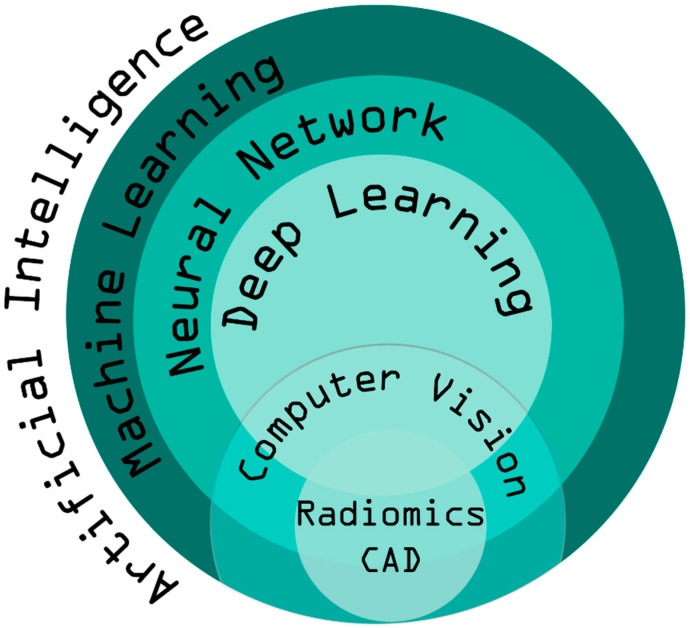
The four terms related to AI and their interrelationship in radiology.

**Figure 2 jpm-12-01429-f002:**
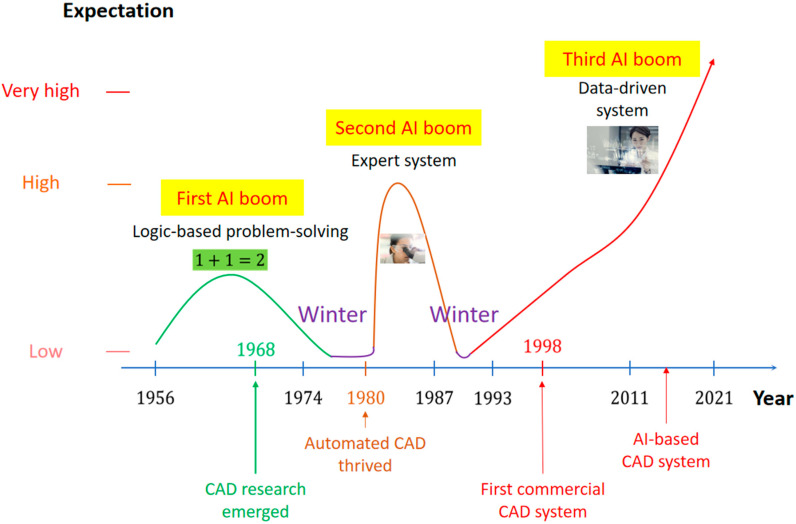
Timeline of AI development with two AI booms in the past and the current third AI boom. Advances in CAD research are intertwined with advances in AI technology.

**Figure 3 jpm-12-01429-f003:**
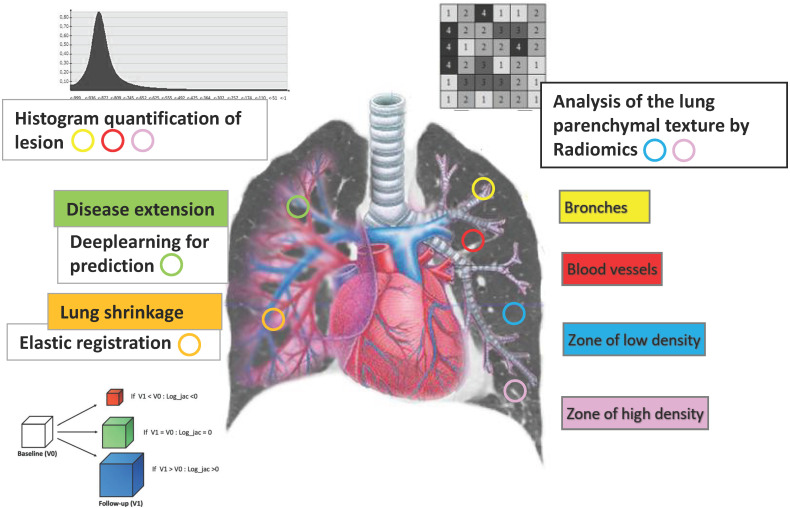
Lung anatomical and physiological parameters explored by artificial intelligence.

## Data Availability

Not applicable.

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
