# Peer review of "How Artificial Intelligence in Imaging Can Better Serve Patients with Bronchial and Parenchymal Lung Diseases?"

_jpm, 2022, doi:10.3390/jpm12091429_

Round 1

Reviewer 1 Report

I recommend a major revision based on the below points.  Please, add a point-to-point response to each comment in your revision:

-          I am not convinced about the novelty of the manuscript. 

-          The abstract is not technical and needs to highlight the research gap clearly.

-          The structure of the paper is vague. The paper needs to be restructured.

-          Don't add heading over heading. Add a few lines related to the detail of a particular section before starting a sub-section.

-          Proofread your paper from a native English speaker. There are many typos and grammar mistakes. 

-          At the end of the Introduction section, add the contributions clearly. See this paper for reference and citation ' A Novel Framework for Prognostic Factors Identification of Malignant Mesothelioma through Association Rule Mining.

-          Section 3 is very broader. It needs to be split into multiple sub-sections.

-          Related work/background/literature review should have a threat to a validity section. At the start of the background section, add a threat to a validity section. In that section, state the search strings and databases that have been explored to find the related work. See the below papers for references and citations 'Performance comparison and current challenges of using machine learning techniques in cybersecurity' and 'A Survey on Machine Learning Techniques for Cyber Security in the Last Decade'.

-          The literature needs to be sub-divided into multiple sub-sections.

-          Add the below papers to your literature: Deep Transfer Learning Approaches in Performance Analysis of Brain Tumor Classification Using MRI Images, Computer-Aided Diagnosis of Coal Workers’ Pneumoconiosis in Chest X-ray Radiographs Using Machine Learning: A Systematic Literature Review, The Impact of Artificial Intelligence and Robotics on the Future Employment Opportunities

-          The authors need to define the limitation of their work i.e., which areas they have not covered in this paper?

-          Add the discussion related to the time complexity factor of AI models. See this paper for reference and citation 'Cyber Threat Detection Using Machine Learning Techniques: A Performance Evaluation Perspective'.

Overall, the paper has many inconsistencies, and the contributions are not clear. The results are not compared with the ground truth properly. Limitations are not provided in their current approach. Future directions are not clearly stated.

I am looking forward to seeing your revised version. 

All the best. 

Reviewer 2 Report

In this review, the authors discussed the role of artificial intelligence in imaging technologies used for bronchial and parenchymal lung diseases diagnosis. The discussion of strengths and limitation of artificial intelligence in thoracic imaging provides the directions for future research. Please see detailed comments below.

1. Page 4, second paragraph last sentence. Please define "PID".

2. In second part of the review, how was artificial intelligence applied in the density measurements, histogram analysis and lung shrinkage detection? Were there any other methods? What are the advantages of artificial intelligence methods over traditional methods?

3. Page 4, paragraph 4. What are the software packages that are available for texture analysis?

4.  Please read through the article carefully and revise the grammar issues and typos. For example, page 4 first paragraph 3rd line, repeated period. Same page third paragraph 2nd sentence, redundant "and" after "pulmonary function". Page 6, second paragraph last line, missing "are" before "able to interpret chest x-rays“。

Reviewer 3 Report

I would like to thank the authors for the good paper, this is one of the best papers I review. Following are my comments:

1- Add a section about machine learning and deep learning in the diagnosis of other lung diseases.

2- Add a table that summarizes the included methods.

3- Add a section that shows the methods of inclusion and exclusion of research in this paper.

4- Add a block diagram for the proposed method of paper construction.

5- Add a section that compares images and sounds in lung disease detection.

6- I suggest including the following research 

A- Pulmonary Diseases Decision Support System Using Deep Learning Approach

B- PneumoniaNet: Automated Detection and Classification of Pediatric Pneumonia Using Chest X-ray Images and CNN Approach

C- Employing Texture Features of Chest X-Ray Images and Machine Learning in COVID-19 Detection and Classification

D-A hybrid deep learning approach towards building an intelligent system for pneumonia detection in chest X-ray images

E- Artificial Intelligence Framework for Efficient Detection and Classification of Pneumonia Using Chest Radiography Images

F- Covid-2019 Detection Using X-Ray Images And Artificial Intelligence Hybrid Systems

Round 2

Reviewer 1 Report

Congratulations